# Mycosynthesis, Characterization, and Mosquitocidal Activity of Silver Nanoparticles Fabricated by *Aspergillus niger* Strain

**DOI:** 10.3390/jof8040396

**Published:** 2022-04-13

**Authors:** Mohamed A. Awad, Ahmed M. Eid, Tarek M. Y. Elsheikh, Zarraq E. Al-Faifi, Nadia Saad, Mahmoud H. Sultan, Samy Selim, Areej A. Al-Khalaf, Amr Fouda

**Affiliations:** 1Department of Zoology and Entomology, Faculty of Science, Al-Azhar University, Nasr City, Cairo 11884, Egypt; mohamed_awad@azhar.edu.eg (M.A.A.); telsheikh64@yahoo.com (T.M.Y.E.); 2Department of Botany and Microbiology, Faculty of Science, Al-Azhar University, Nasr City, Cairo 11884, Egypt; prof.mahmoud@azhar.edu.eg; 3Center for Environmental Research and Studies, Jazan University, P.O. Box 2097, Jazan 42145, Saudi Arabia; zalfifi@jazanu.edu.sa; 4Department of Mathematics, Faculty of Science, Helwan University, Cairo 11795, Egypt; nadia_saad@science.helwan.edu.eg; 5Department of Clinical Laboratory Sciences, College of Applied Medical Sciences, Jouf University, P.O. Box 72388, Sakaka 72341, Saudi Arabia; sabdulsalam@ju.edu.sa; 6Biology Department, College of Science, Princess Nourah Bint Abdulrahman University, P.O. Box 84428, Riyadh 11671, Saudi Arabia; aaalkhalaf@pnu.edu.sa

**Keywords:** biosynthesis, silver nanoparticles, biocontrol, larvicidal, smoke toxicity, ovicidal

## Abstract

Herein, silver nanoparticles (Ag-NPs) were synthesized using an environmentally friendly approach by harnessing the metabolites of *Aspergillus niger* F2. The successful formation of Ag-NPs was checked by a color change to yellowish-brown, followed by UV-Vis spectroscopy, Fourier transforms infrared (FT-IR), Transmission electron microscopy (TEM), and X-ray diffraction (XRD). Data showed the successful formation of crystalline Ag-NPs with a spherical shape at the maximum surface plasmon resonance of 420 nm with a size range of 3–13 nm. The Ag-NPs showed high toxicity against I, II, III, and IV instar larvae and pupae of *Aedes aegypti* with LC50 and LC90 values of 12.4–22.9 ppm and 22.4–41.4 ppm, respectively under laboratory conditions. The field assay exhibited the highest reduction in larval density due to treatment with Ag-NPs (10× LC50) with values of 59.6%, 74.7%, and 100% after 24, 48, and 72 h, respectively. The exposure of *A. aegypti* adults to the vapor of burning Ag-NPs-based coils caused a reduction of unfed individuals with a percentage of 81.6 ± 0.5% compared with the positive control, pyrethrin-based coils (86.1 ± 1.1%). The ovicidal activity of biosynthesized Ag-NPs caused the hatching of the eggs with percentages of 50.1 ± 0.9, 33.5 ± 1.1, 22.9 ± 1.1, and 13.7 ± 1.2% for concentrations of 5, 10, 15, and 20 ppm, whereas Ag-NPs at a concentration of 25 and 30 ppm caused complete egg mortality (100%). The obtained data confirmed the applicability of biosynthesized Ag-NPs to the biocontrol of *A. aegypti* at low concentrations.

## 1. Introduction

Nanotechnology deals with the production of new particles at a nano-scale (1–100 nm) [1]. Nanotechnology science paved the way for discovering active compounds that can be incorporated into various fields such as sensors, magnetic devices, the textile industry, wastewater treatment, heavy metal removal, drug delivery, optoelectronics, the agricultural sector, biomedical (antimicrobial, antitumor, cytotoxicity, and cosmetics), and parasitology [2,3,4,5]. These activities are due to the unique physical, chemical, and structural nanoparticle (NPs) properties such as sizes, shapes, the charge of the surface, stability, compatibles, and the proportion between small particle size to a large surface area [6,7]. Nanoparticles are synthesized by chemical, physical and biological methods. The main disadvantages of the chemical and physical methods are expensive, producing hazardous substances, and requiring harsh conditions (such as temperature and pressure) during synthesis [8,9]. These disadvantages paved the way to green or biological synthesis using different biological beings such as plants and microorganisms (bacteria, fungi, actinomycetes, yeast, and algae) [10,11,12]. Fungi are gaining more attention than other microorganisms for fabricating different metal and metal oxide NPs [13,14]. This activity can be attributed to the ability of fungi to have high heavy metal tolerance, easy scale-up, high biomass production, easy handling, low toxicity, and production of various quantities of secondary metabolites that increase the stability of NPs [15,16]. Various metals, metal oxide NPs, and nanocomposites were fabricated by fungi such as Ag, Au, Se, ZnO, CuO, MgO, CuO/ZnO nanocomposites, etc. [3,8,17,18]. 

Mosquitoes are one of the most abundant vectors for various deadly human and animal pathogens such as viruses, protozoa, and bacteria [19]. A variety of deadly diseases such as dengue, yellow fever, filariasis, chikungunya, malaria, Zika virus, and West Nile virus are considered mosquito-borne diseases [20]. According to the database of the World Health Organization, approximately 50 to 100 million dengue fever cases appear every year worldwide [21]. *Aedes aegypti* is a common mosquito species in the tropic as well as subtropic countries and is characterized by its ability to transmit several virus-causing diseases such as Zika, dengue, chikungunya, and yellow fever [22]. Interestingly, the viral diseases caused by *A. aegypti* do not have effective vaccines except for yellow fever, which has had an effective vaccine since the 1940s [23]. Therefore, successful protection from *A. aegypti*-caused diseases is accomplished by preventing the spread of the vector. 

The *Aedes* spp. control was achieved by synthetic pesticides such as organophosphates, organochlorines, dichloro-diphenyl-trichloroethane (DDT), carbamates, and pyrethroid [24,25]. The continuous usage of these chemicals leads to increased mosquito resistance, negative impacts on soil fertility, toxic groundwater, and adverse effects on the surrounding ecosystem [2,26]. Therefore, the main target for researchers is to discover alternative eco-friendly, cost-effective, and safe tools to control and prevent the spread of insect vectors to overcome the negative impacts of chemical insecticides. 

Concerning the control of mosquito vectors, green synthesized NPs are used due to their eco-friendliness, rapid effects, cost-effectiveness, high stability, and absence of negative impacts on public health compared to chemical insecticides [27]. Recently, selenium nanoparticles (Se-NPs) synthesized by *Penicillium chrysogenum* showed significant molluscicide toxicity against *Biomphlaria alexandrina* snails at a concnetration of 5.9 mg L^−1^ after 96 h. Additionally, it showed cercaricidal and miracidicidal effect on *Schistosoma mansoni* [28]. In our recent study, magnesium oxide NPs (MgO-NPs) fabricated by *Cystoseira crinita* showed high efficacy as larvicidal, pupicidal, and repellence activity for *Musca domestica* [29]. Silver nanoparticles (Ag-NPs) are considered one of the most critical metal NPs which can be integrated into different biomedical and biotechnological applications. Recently, Ag-NPs were used to control mosquito vectors such as *Culex quinquefasciatus*, *Anopheles stephensi*, and *Aedes aegypti* [2,30].

Therefore, the efficiency of metabolites secreted by fungal strains for the synthesis of Ag-NPs and the use of the final product to control the *Aedes aegypti* mosquito was investigated. The biomass filtrate of *Aspergillus niger* F2 that contains a variety of metabolites was used as a biocatalyst to form Ag-NPs as an eco-friendly approach. The physicochemical characterizations of biosynthesized NP were accomplished by the color change of fungal biomass filtrate, UV-Vis spectrophotometry, Fourier transforms infrared (FT-IR), transmission electron microscopy (TEM), and X-ray diffraction (XRD). The mechanisms of synthesized nano-silver to control *A. aegypti* including larvicidal activity against different instar larvae under laboratory and field conditions, pupicidal, smoke toxicity, and ovicidal activity were investigated. 

## 2. Materials and Methods

### 2.1. Fungal Strain

The biological synthesis of Ag-NPs was done using the fungal strain *Aspergillus niger* F2 which was previously isolated from the archeological manuscript [31]. This manuscript was dated back to 1677 A.D. and collected from Al-Azhar Library, Cairo, Egypt. The identification of the selected fungal strain was achieved by cultural characteristics, microscopic examination, and molecular identification by sequencing of the internal transcribed spacer (ITS) gene [32]. The obtained gene sequence was deposited in GenBank under accession number MK452259.

### 2.2. Mycosynthesis of Ag-NPs

Two disks (1.0 cm in diameter) of three-day-old *A. niger* F2 culture were inoculated in potato dextrose (PD) broth media and incubated for 120 h at 30 ± 2 °C. At the end of the incubation period, the inoculated PD was filtered with Whatman (No. 1) filter paper to collect the fungal biomass which was washed thrice with sterilized distilled H_2_O to remove any adhering medium components. After that, approximately 10 g of collected fungal biomass was mixed with 100 mL of distilled H_2_O and incubated for 24 h under shaking conditions (150 rpm). The previous mixture was centrifuged at 5000 rpm for 10 min to collect the supernatant (fungal biomass filtrate) which was used as a biocatalyst for eco-friendly synthesis of Ag-NPs as follows: 16.9 µg of AgNO_3_ (metal precursor) was mixed with 100 mL of fungal biomass filtrate to get a final concentration of 1 mM, adjust the pH of the mixture at 8 using 1 M NaOH which added drop-wisely under stirring condition, and incubated for 24 h at 35 ± 2 °C. The successful synthesis of Ag-NPs was checked through a color change of fungal biomass filtrate from colorless to yellowish-brown. The resultant was collected, washed thrice with distilled H_2_O, and subjected to oven-dry at 150 °C overnight [33]. The negative control with either AgNO_3_ aqueous solution or fungal biomass filtrate without AgNO_3_ was running with the experiment under the same condition.

### 2.3. Ag-NPs Characterization

The synthesis of Ag-NPs was checked by measuring the absorbance properties of a synthesized aqueous solution by UV-Vis spectroscopy (JENWAY 6305, Spectrophotometer, 230 V/50 Hz, Staffordshire, UK) at a different wavelength in the range of 300 to 600 nm with 10 nm intervals [34]. The role of various functional groups present in fungal biomass filtrate in the reduction and stabilizing Ag-NPs was analyzed by Fourier-transformed infrared (FT-IR) spectroscopy (Cary 630 FTIR model, Tokyo, Japan). The synthesized Ag-NPs (0.3 g) were mixed with potassium bromide (KBr) and formed a disk under high pressure followed by scanning in a range of 400 to 4000 cm^−1^ [35]. The sizes and shapes of biosynthesized Ag-NPs were detected using Transmission Electron microscopy (TEM) (TEM-JEOL 1010, Tokyo, Japan). A drop of Ag-NPs solution was added to the TEM grid (carbon-copper gride) till complete adsorption. The excess of the Ag-NPs solution on the TEM grid was removed before being analyzed [36]. The crystalline nature of synthesized Ag-NPs was assessed by X-ray Diffraction Analysis (X′ Pert Pro, Philips, Eindhoven, The Netherlands). The sample was scanned at 2θ values of 4 to 80. The XRD operates at 30 mA and 40 KV with a radiation source of Cu Ka (λ = 1.54 Å). Based on XRD analysis, the average Ag-NPs size was calculated by Debye–Scherrer equation as follows [37]:(1)$$D=\frac{0.9\times 1.54}{\beta \cos\theta }$$
where D is the mean particle size, 0.9 is the Scherrer’s constant, 1.54 is the X-ray wavelength, β is half of the maximum intensity, and θ is Bragg’s angle. 

### 2.4. Mosquito Culture

The eggs of *Aedes aegypti* were purchased from the Medical Entomology Institute, Giza, Egypt, and transferred immediately to the Animal House Institute, Mosquito Laboratory, Zoology Department, Faculty of Science, Al-Azhar University, Cairo, Egypt. The collected eggs were put into a plastic cup containing 0.5 L of tap water to hatch at optimum conditions (27 ± 2 °C, 75–85% humidity, and the light–dark photoperiod was 14:10). The hatched larvae were fed on yeast hydrolyzed–dog biscuits (1:3 *w/w*). The released pupae were put into a plastic cup filled with 0.5 L dechlorinated water followed by transfer to a chiffon cage (90 × 90 × 90 cm^3^) until adults were released which fed on a 10% (*v*/*v*) sucrose solution [38]. The released larvae and pupae were collected for the next step which studied the toxicity of synthesized Ag-NPs.

### 2.5. Laboratory Larvicidal/Pupicidal Toxicity of Ag-NPs

The toxicity of biosynthesized Ag-NPs against various larval instars (I, II, III, and IV instars) and pupae of *A. aegypti* were assessed through the preparation of suspension solution in distilled H_2_O at different concentrations of 5, 10, 15, 20, 25, and 30 ppm. The differences between insect instars were defined by alterations in the body proportions, growth patterns, colors, head width, and changes in the number of body segments [39]. Briefly, 25 larvae or pupae were incubated for 24 h in a glass cup containing 0.5 L dechlorinated water supplemented with the above-mentioned Ag-NPs concentration and 0.5 mg of larvae food. The experiment was conducted for each instar larvae and pupae with Ag-NPs concentration separately and then repeated three times. The control where larvae and pupa were in dechlorinated water without Ag-NPs was running with the experiment under the same conditions. After 24 h, the mortality percentages were calculated according to the following equation [7]: (2)$$\mathrm{Mortality}\;\mathrm{percentages}\;(\%)=\frac{\mathrm{Number}\;\mathrm{of}\;\mathrm{dead}\;\mathrm{individuals}}{\mathrm{Number}\;\mathrm{of}\;\mathrm{treated}\;\mathrm{individuals}}\times 100$$

### 2.6. Field Larvicidal Bioassay 

The toxicity of Ag-NPs against 3rd and 4th instar larvae of *A. aegypti* under field conditions was assessed using a knapsack sprayer in six water reservoirs at the Animal House Institute, Mosquito Laboratory, Faculty of Science, Al-Azhar University, Cairo, Egypt. The density of larvae before treatment was checked, whereas the toxicity after Ag-NPs treatment was investigated at 24, 48, and 72 h by a larval dipper. The experiment was repeated six times under the same field conditions of 80 ± 5% humidity at 28 ± 2 °C. The required Ag-NPs as a mosquitocidal agent were calculated according to the volume and surface area of reservoirs, which were prepared as 10× LC50 values as mentioned above [40]. The reduction percentages in the larvae density were calculated using the following formula [41]: (3)$$\mathrm{Reduction}\;\mathrm{percentages}\;(\%)=\frac{C-T}{C}\times 100$$
where C is the total number of individuals in control and T is the total number of individuals in treatment.

### 2.7. Smoke Toxicity Assay

The smoke toxicity of Ag-NPs against adults of *A. aegypti* was achieved in a glass chamber (60 cm × 40 cm × 35 cm). In this method, the burning coils were composed of 1 g of biosynthesized Ag-NPs mixed with 0.5 g of binding materials (sawdust) and 0.5 g of burning materials (coconut shell powder). The components were mixed well with distilled H_2_O to form a semisolid paste which was used to form a mosquito coil that remained in the shade to dry. Negative control (sawdust–coconut shell powder (0.5:0.5 *w*/*w*), mixed well with distilled H_2_O to form a semisolid paste which remains in shade to dry) and positive control (the same previous components and mixed with pyrethrin instead of Ag-NPs) were running with the experiment under the same conditions. 

In the experiment, approximately 100 adult female mosquitoes (blood-starved for an average age of three to four days) were released in a glass chamber containing a sucrose solution (10%). Additionally, a pigeon with a shaved belly was tied to the side of the chamber. This study was approved by the Ethical Committee of the Animal House Institute, Cairo, Egypt. The released adult mosquitos were exposed to the vapor of the burning coil for one hour followed by counting the number of fed and unfed (alive and dead) mosquitos. The protection of the pigeon against the bites of *A. aegypti* due to smoke from Ag-NPs was calculated as follows [42]: (4)$$\begin{array}{l} \mathrm{Protection}\;\%=\\ \frac{\mathrm{number}\;\mathrm{of}\;\mathrm{unfed}\;\mathrm{mosquitos}\;\mathrm{in}\;\mathrm{treatment}-\mathrm{number}\;\mathrm{of}\;\mathrm{unfed}\;\mathrm{mosquitos}\;\mathrm{in}\;\mathrm{negative}\mathrm{control}}{\mathrm{number}\;\mathrm{of}\;\mathrm{treated}\;\mathrm{mosquitos}}\times 100 \end{array}$$

### 2.8. Ovicidal Activity

The mosquito ovicidal assay with Ag-NP treatment was achieved according to the method of Su and Mulla [43]. In this assay, the eggs of *A. aegypti* were collected and placed in ovitraps (such as Petri dishes with a diameter of 60 mm containing filter papers and 100 mL distilled H_2_O) and placed into the mosquito cages for 48 h. The photomicroscope was used to check the eggs loaded on the filter paper. After that, out of seven glass cups were put in the mosquito cage, six cups were filled with water supplemented with Ag-NPs concentrations (5, 10, 15, 20, 25, and 30 ppm), and the seventh cup was filled with water only (as a control). Approximately 100 *A. aegypti* eggs were put in each glass cup. The experiment was repeated three times for each treatment. After 48 h, the eggs from each treatment were transferred to a second cup containing distilled H_2_O to investigate the % eggs hatched under a microscope. According to non-hatched eggs (detected by unopen opercula), the percentages of eggs hatched were calculated according to the following formula: (5)$$\%\;\mathrm{eggs}\;\mathrm{hatched}=\frac{\mathrm{number}\;\mathrm{of}\;\mathrm{hatched}\;\mathrm{larvae}}{\mathrm{Total}\mathrm{number}\;\mathrm{of}\;\mathrm{tested}\;\mathrm{eggs}}\times 100$$

### 2.9. Data Analysis

The data collected in the current study were analyzed using the SPSS (version 16.0). Data of acute toxicity obtained from Laboratory assays were transformed into arcsine/proportion values followed by two-way ANOVA analyses with two factors (i.e., dosage and mosquito instar). Furthermore, insect pest mortality data from laboratory assays were analyzed using probit analysis, with LC50 and LC90 calculated using Finney’s method [44]. A two-way ANOVA with two factors was used to analyze the larval density data of *A. aegypti* obtained from field assays. In the smoke coil toxicity experiment, the number of fed and unfed mosquitos was analyzed using a one-way ANOVA with the treatment as the factor (i.e., the coil). Moreover, one-way ANOVA was used to analyze ovicidal data that had been transformed into arcsine proportion values. A posteriori multiple comparisons were done using Tukey’s range tests at *p ≤* 0.05. All results are the means of three to five independent replicates, as specified above. 

## 3. Results and Discussion

### 3.1. Mycosynthesis of Ag-NPs

In the current study, the Ag-NPs were synthesized extracellularly, this mode facilitates the purification method as compared with intracellular synthesis. The fungal strain F2 was selected in the current study due to their uncommon habitat (deteriorated archaeological manuscript) and hence predict their high activity via secretion of various enzymes and other secondary metabolites. Therefore, the biomass filtrate containing various secondary metabolites of *Aspergillus niger* F2 was used as a reducing agent for the silver ions to form silver nanoparticles (Ag-NPs) as well as being used as a capping/stabilizing agent for a new nanostructure. The pH of the reaction solution is considered one of the critical parameters during NP synthesis [45]. Herein, the alkaline condition (pH 8) of the synthesis solution was preferred over neutral and acidic conditions. This phenomenon can be attributed to the alkaline conditions enhancing the reducing activity of different functional groups that exist in the fungal biomass filtrate as well as preventing the aggregation or agglomeration of NPs [46]. Moreover, the alkaline medium can help in capping and stabilizing of NPs via reacting with amino acids and amino groups that exist on the surface of proteins [47]. The efficacy of biomass filtrate to form Ag-NPs was checked first by the change of its color to yellowish-brown because of exciting the surface plasmon vibrations [48]. Compatible with our study, *Aspergillus niger* was utilized as a biocatalyst for the biological synthesis of Ag-NPs as mentioned in the various published studies [36,45,49]. The biosynthesis of NPs using different biological sources (i.e., bacteria, fungi, actinomycetes, algae, and plants) have advantages over chemical and physical methods because of the eco-friendly, rapid, low cost, and biocompatible methods [7,50]. Recently, NPs have been fabricated by different fungal strains to itegrate into a wide range of applications [28,51,52]. The reduction process of silver ions to the nanoscale can be achieved due to the liberated electrons from the reduction of NO_3_ to NO_2_ by the action of metabolites present in fungal biomass filtrate, and hence, the color intensity is directly related to the number of liberated electrons and surface plasmon resonance (SPR) which varied according to the size of NPs [53,54]. 

### 3.2. Ag-NPs Characterizations

#### 3.2.1. UV-Vis Spectroscopy

The color intensity of fungal-mediated biosynthesis of Ag-NPs was checked by measuring the absorbance at various wavelengths in the range of 300 to 600 nm to detect the maximum SPR. As shown, the maximum peak of Ag-NPs was observed at 420 nm (Figure 1), which confirms the formation of Ag-NPs [55]. The spherical shape of biosynthesized Ag-NPs is usually related to the SPR peak especially those observed at a wavelength of 410–420 nm [56]. Previously published studies reported that the optimum SPR absorption peak for biologically synthesized Ag-NPs was in the range of 400–460 nm [33,57]. Compatible with our study, Wang and co-author reported that the optimum SPR peak for Ag-NPs synthesized by harnessing metabolites of *Aspergillus sydowii* was at 420 nm [34]. 

#### 3.2.2. Fourier Transform Infrared (FT-IR) Analysis

The various functional groups present in biomass filtrate and their roles in the reduction and capping/stabilizing of mycosynthesized Ag-NPs were investigated by FT-IR. As shown, there are six peaks of fungal biomass filtrate at a wavenumber of 638, 1030, 1630, 2060, 2360, and 3430 cm^−1^ (Figure 2A). The broad peak at 538 cm^−1^ corresponds to C–Br of halo compounds, whereas the medium peak at 1030 cm^−1^ signifies the stretching C–O, or CN, or bending C–H of primary amines [58,59]. The strong peak at 1630 cm^−1^ is related to C=O of polysaccharide moieties [29]. The appearance of a broad peak at 2060 cm^−1^ can be attributed to N=C=S of isothiocyanate. The weak peak at 2360 cm^−1^ signifies CO_2_, whereas the broad strong peak at 3430 cm^−1^ could be related to either the O–H or N–H group of amino acids that exist in fungal biomass filtrate [59,60]. 

On the other hand, the FT-IR spectra of mycosynthesized Ag-NPs showed eight peaks at a wavenumber of 528, 1110, 1325, 1450, 1630, 2344, 2900, and 3290 cm^−1^ (Figure 2B). The strong, broad peak at 3290 signified the O–H stretching of carboxylic acid [61], whereas the peak at 2900 cm^−1^ can be attributed to the C–H stretching of aliphatic hydrocarbons [62,63]. The appearance of the peak at 2344 cm^−1^ is related to the O=C=O stretching of CO_2_ that is adsorbed onto the surface of proteins [59,64]. On the other hand, the medium peak at 1630 cm^−1^ signified the C═O of polysaccharide moieties, whereas the peak at 1450 cm^−1^ is related to the C–H bending of the methyl group. The peak at 1325 cm^−1^ is related to the C–N stretching of aromatic amines, whereas the strong peak at 1110 cm^−1^ corresponds to the C–O stretching vibration of the carbohydrate ring [59,63]. Finally, the peak at 528 cm^−1^ represents the vibration of C–Br stretching alkyl halides [65]. The obtained FT-IR results confirm the presence of various bioactive molecules such as carbohydrates, alkenes, carboxylate, and amino acids that have been reported previously as a potential reducing agent for the biosynthesis of metal and metal oxide NPs [10,66].

#### 3.2.3. Transmission Electron Microscopy (TEM) 

The size and shape of Ag-NPs fabricated by *A. niger* F2 were investigated by TEM analysis. As shown, the as-formed NPs shape was spherical and well-dispersed with sizes in the range of 2–13 nm and an average size of 8.72 ± 2.21 nm (Figure 3A,B). According to size distribution, it can be concluded that the size of the majority of fabricated Ag-NPs was less than 10 nm. Similarly, the biomass filtrate of *A. niger* strain NRC1731 was used to fabricate spherical Ag-NPs with sizes ranging between 3 nm to 20 nm [45]. Additionally, Li et al. successfully formed well-dispersed and spherical Ag-NPs with an average size of 4.3 nm through harnessing metabolites of *Aspergillus terreus* [57]. The various applications of biosynthesized Ag-NPs were highly dependent on several parameters such as chemical compositions, sizes, shapes, and crystallographic structure [67]. The activity of as-formed metal NPs was closely related to their size, meaning the smaller size predicts high activity as previously reported [68]. For example, the activity of NPs formed by an aqueous extract of garlic with the size of 21–40 nm was higher than those with sizes of 41–50 nm [69]. Additionally, the antibacterial activity of small size Ag-NP was better than those recorded for big particle sizes as reported [70]. The obtained sizes can predict high activity for the fungal-mediated synthesis of Ag-NPs. 

#### 3.2.4. X-ray Diffraction Pattern

The crystalline and amorphous nature of mycosynthesized Ag-NPs was analyzed by XRD in the 2θ range of 10–80 (Figure 4). As shown, the XRD spectra contain four intense peaks represented by (111), (200), (220), and (311) at 2θ values of 38.4°, 44.4°, 64.3°, and 77.4°, respectively, which are indexed for a face-centered-cubic (FCC) structure of biosynthesized Ag-NPs [71]. The XRD spectra are completely compatible with the JCPDS (Joint Committee on Powder Diffraction Standards) card (No. 04-0783) for the crystalline nature of Ag-NPs [57]. Thomas et al. reported the absence of other diffraction peaks in the XRD chart because of the successful crystallization of metabolites that are used to coat and stabilize Ag-NPs [72]. The obtained XRD results reflect those reported by Wang et al., who successfully fabricated the crystalline nature of Ag-NPs at 2θ values of 38.2°, 44.4°, 64.6°, and 77.8° by using a biomass filtrate of *Aspergillus sydowii* [34]. The average crystal size of synthesized Ag-NPs can be calculated using XRD analysis by the Debye–Scherrer equation which was 20 nm. TEM and XRD analysis revealed that the mycosynthesized Ag-NPs have a uniform morphological shape which was spherical and small average size. 

### 3.3. Larvicidal/Pupicidal Toxicity of Ag-NPs under Laboratory and Field Conditions

Herein, the fungal mediated synthesis of Ag-NPs showed high efficacy as larvicide and pupicide for *A. aegypti* at various concentrations of 5, 10, 15, 20, 25, and 30 ppm under laboratory conditions. Data analysis showed that the toxicity of Ag-NPs against various instar larvae and pupae of *A. aegypti* was dose-dependent, and this phenomenon was in agreement with various published studies [41,73,74]. Recently, Ag-NPs fabricated by biological route have been showing high toxicity against a variety of mosquitos [65,75]. The LC50 values (that kill 50% of individuals) and LC90 values (that kill 90% of individuals) were 12.4, 13.6, 15.04, 20.9, and 22.9 ppm and 22.4, 24.6, 27.1, 37.6, and 41.4 ppm for I, II, III, and IV instar larvae and pupae, respectively (Table 1). Compatible with the obtained results, the Ag-NPs synthesized by an aqueous extract of *Suaeda maritima* exhibit high toxicity against first instar larvae and pupa of *A. aegypti* with LC50 values of 8.7 and 17.9 ppm, respectively [73]. Additionally, the Ag-NPs fabricated by *Turbinaria ornata* showed high activity against *Culex quinquefasciatus*, *Anopheles stephensi*, and *A. aegypti* with LC50 values of 1.5, 1.13, and 0.74 µg mL^−1^ [75]. Data reported by Jinu et al. showed that the Ag-NPs synthesized by an aqueous extract of *Strychnos nuxvomica* or *Cleistanthus collinus* were highly toxic for larvae of *Anopheles stephensi* and *A. aegypti* with IC50 values of 8.8 and 7.8 ppm and 11.1 and 11.4 ppm, respectively [11].

Under field conditions, the reduction in larval density was time-dependent. Data analysis showed that the treatment of *A. aegypti* larvae in a water storage reservoir with Ag-NPs (10× LC50) fabricated by fungal strain *A. niger* F2 caused a complete reduction (100%) after 72 h (Table 2). The obtained data are in agreement with published studies. For instance, the treatment of larvae of *A. aegypti* with Ag-NPs synthesized by leaf extract of *Phyllanthus niruri* led to a complete reduction in larval density under field assay after 72 h [42]. Additionally, the larval density of *Anopheles stephensi* was reduced by a percentage of 97.7% due to treatment with Ag-NPs fabricated by *Aloe vera* in a water storage reservoir [41]. In the current study, the reduction in larval density of *A. aegypti* due to a single treatment with Ag-NPs was achieved with percentages of 59.6% and 74.7% after 24 and 48 h, respectively (Table 2). The difference between Ag-NPs activity in the laboratory and filed assay can be attributed to various reasons such as no proper distribution of the NP across the containers, aggregation of NP at a large scale, and uncontrollable environmental field conditions. 

The high toxicity of Ag-NPs synthesized by a biomass filtrate of *A. niger* F2 can be attributed to their small size (2–13 nm), which is easily adsorbed through the cuticle or digestive tract of an insect and hence interferes with the most physiological processes, ultimately leading to cell death [73]. Moreover, once Ag-NPs enter the insect cells, they can be react with various cellular macromolecules such as amino acids, proteins, and DNA, and eventually lead to cell death [76]. Baskar et al. reported that the activity of Ag-NPs toward various insects was related to their efficacy in blocking various metabolic activities through binding with hormones related to the synthesis of proteins [77]. The toxicity of Ag-NPs against pupae and adults can be attributed to the deformed abnormal morphology or swelling of the pupal integument with the exoskeleton shrinking during death as previously reported [2].

### 3.4. Smoke Toxicity Assay

Table 3 shows the smoke toxicity assay of Ag-NPs-based coils against the adults of *A. aegypti.* As shown, after exposure to the vapor of burning Ag-NPs-based coils, the mean percent of unfed *A. aegypti* was 81.6 ± 0.5% as compared to the percentages of negative control and pyrethrin-based coil (positive control) which were 23.9 ± 1.2% and 86.1 ± 1.1%, respectively. Analysis of variance revealed a slight change between the mortality percentages of a pyrethrin-based coil and Ag-NPs-based coils (Table 3). Based on these results, it can be concluded that the Ag-NPs-based coils can be used instead of pyrethrin-based coils as an eco-friendly approach to control the adults of *A. aegypti.* To the best of our knowledge, this is the first report to investigate the smoke toxicity of fungal mediated biosynthesis of Ag-NPs-based coils against the adults of *A. aegypti.* According to the obtained data, synthesized Ag-NPs-based coils were more efficient compared to botanical-based coils. The mortality percentages caused by exposure to the vapor of coils formed from the leaves, stems, and roots of *Suaeda maritima* were 86.6%, 79.8%, and 72.7%, respectively [73]. Moreover, the mean percentages of unfed *A. aegypti* mosquitos due to treatment with the stems, leaves, and roots of *Phyllanthus niruri*-based coils were 40%, 58%, and 61%, respectively [42]. The smoke toxicity of Ag-NPs-based coils can be attributed to their toxic effects on the central nervous system of mosquitos upon exposure to the vapor released from burning coils [78]. Moreover, the vapor released from coil burning can be causing acute irritation to the upper respiratory tracts [79]. The coil burning produces unhealthy air conditions that direct entry to internal organs, ultimately severe organs paralysis, producing smaller larvae, and reducing the fecundity [80]. 

### 3.5. Ovicidal Activity

The efficacy of various concentrations of *A. niger* mediated biosynthesis of Ag-NPs (5, 10, 15, 20, 25, and 30 ppm) on the egg hatchability of *A. aegypti* was investigated. Data analysis showed that the egg hatchability of *A. aegypti* was Ag-NPs dose-dependent as reported previously [30,73]. As shown in Figure 5, the hatchability of the eggs of *A. aegypti* was completely eliminated with percentages of 100% after treatment with high concentrations (25 and 30 ppm) of Ag-NPs as compared with the control that showed hatchability percentages of 89.01 ± 0.42%. Moreover, the low concentration of biosynthesized Ag-NPs exhibits toxicity toward egg hatchability. Therefore, the egg hatchability after treatment with 5, 10, 15, and 20 ppm of Ag-NPs were 50.1 ± 0.9, 33.5 ± 1.1, 22.9 ± 1.1, and 13.7 ± 1.2%, respectively. The obtained data agree with those reported by Suresh et al., who reported that the egg hatchability percentages of *A. aegypti* after treatment with 5, 10, and 15 ppm of Ag-NPs were 53.0 ± 1.6, 36.6 ± 1.1, and 25.2 ± 1.3%, respectively, whereas the concentration of 20 and 25 ppm caused the complete reduction (100%) of egg hatchability [73]. The literature survey showed that the nanomaterials were shown ovicidal activity at low concentrations as compared to botanical extract. For instance, 100% egg mortality of *A. aegypti* was attained after treatment with an aqueous leaf extract of *Dicranopteris linearis* and Ag-NPs fabricated by the same plant at a concentration of 300 ppm and 25 ppm, respectively [81]. 

## 4. Conclusions

In the current study, Ag-NPs were synthesized by an environmentally friendly approach using a biomass filtrate containing metabolites of *A. niger* F2. Based on the characterization study, the FT-IR analysis revealed the role of various metabolites present in fungal biomass filtrate in the reduction and stabilization of Ag-NPs. Moreover, the crystalline nature and spherical shape of Ag-NPs with sizes ranging between 3–13 nm were confirmed by XRD and TEM analysis. Under laboratory conditions, the LC50 and LC90 for Ag-NPs against *A*. *aegypti* were in the range of 12.4–22.9 ppm and 22.4–41.4 ppm, respectively. In a field assay, Ag-NPs showed high reduction percentages of larval intensity with values of 100% after 72 h. Moreover, the vapor released from the burning of Ag-NPs-based coils causes a reduction percentage of 81.6 ± 0.5% in *A. aegypti* adults compared to the pyrethrin-based coils (86.1 ± 1.1%). Additionally, the Ag-NPs concentration of 25 and 30 ppm caused the complete egg mortality of *A. aegypti.* The obtained data confirmed the efficacy of Ag-NPs synthesized by the eco-friendly approach to be used as an insecticide instead of a chemical formulation that has negative impacts on the environmental eco-system and human public health. 

## Figures and Tables

**Figure 1 jof-08-00396-f001:**
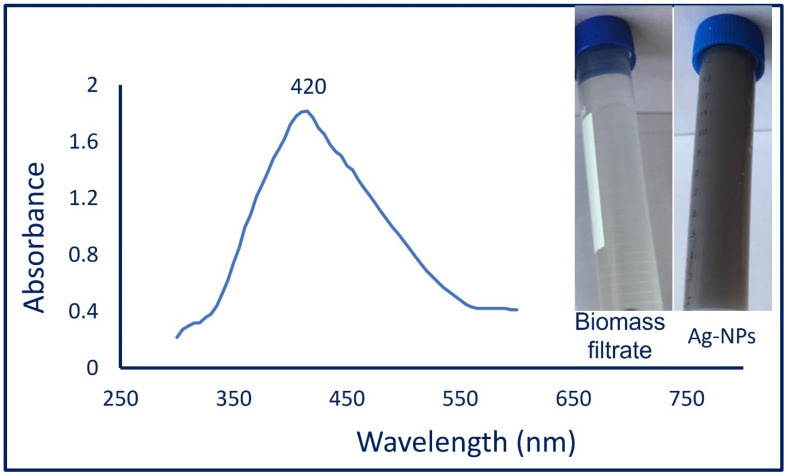
Color change and UV-Vis spectroscopy analysis of mycosynthesized Ag-NPs at wavelengths of 300–600 nm.

**Figure 2 jof-08-00396-f002:**
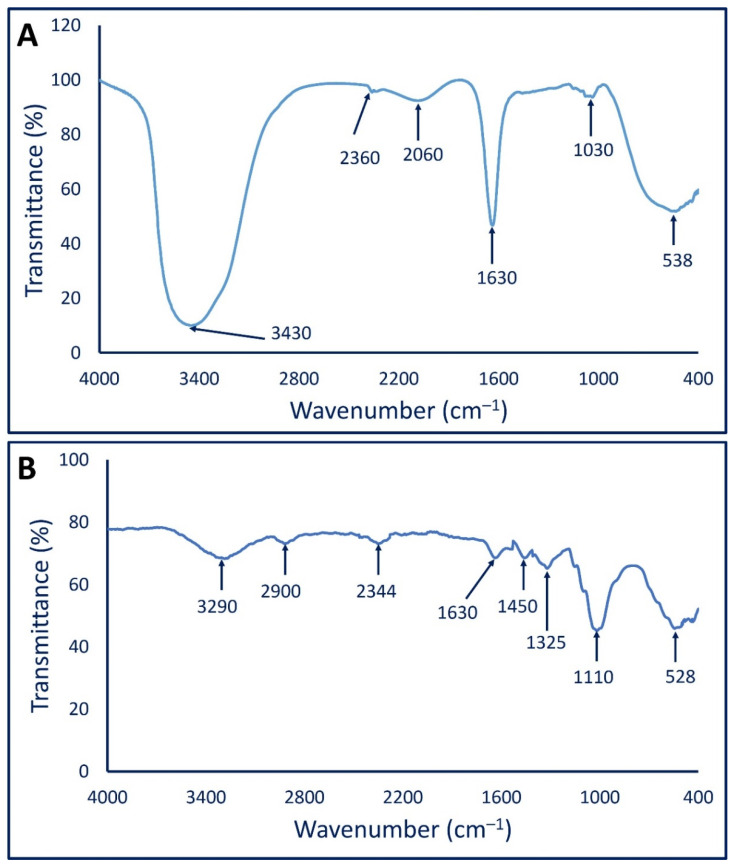
FT-IR spectra of fungal biomass filtrate (**A**) and Ag-NPs fabricated by *Aspergillus niger* strain F2 (**B**).

**Figure 3 jof-08-00396-f003:**
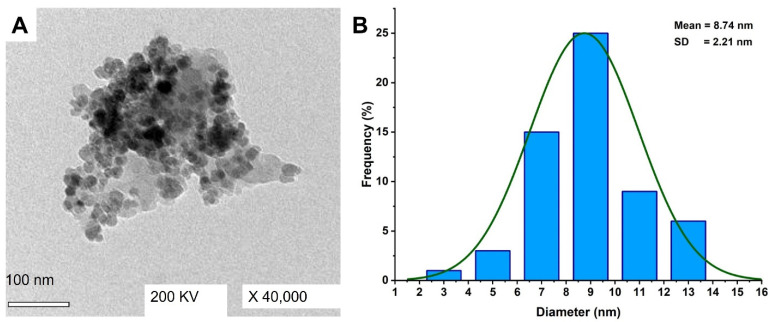
Characterization of Ag-NPs formed by biomass filtrate of *A. niger* strain F2. (**A**) is the TEM image, and (**B**) is the size distributions based on the TEM image.

**Figure 4 jof-08-00396-f004:**
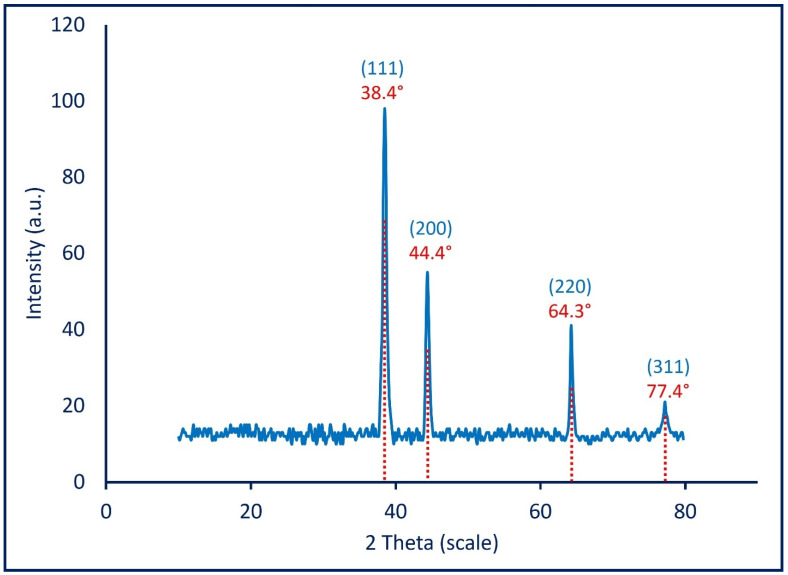
The XRD pattern of mycosynthesized Ag-NPs shows the crystalline nature.

**Figure 5 jof-08-00396-f005:**
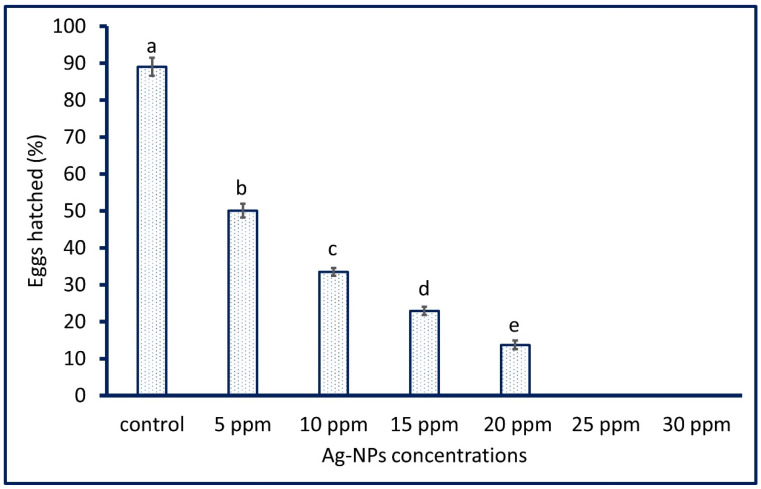
The eggs hatched percentages (%) of *A. aegypti* after treatment by mycosynthesized Ag-NPs at various concentrations. Data are represented by the mean ± SD (*n* = 5). Different letters between the column are significantly different (*p* < 0.05).

**Table 1 jof-08-00396-t001:** The toxicity of Ag-NPs fabricated by *A. niger* strain F2 against various instar larvae and pupae of *A. aegypti*.

Target	LC50 (LC90) (ppm)	95 % Confidence Limit LC50 (LC90)	
		LCL	UCL
I instar	12.4 (22.4)	10.01 (20.3)	16.03 (25.5)
II instar	13.6 (24.6)	11.9 (25.2)	14.8 (27.1)
III instar	15.04 (27.1)	12.3 (21.1)	16.03 (29.8)
IV instar	20.9 (37.6)	18.4 (30.6)	24.4 (39.1)
Pupa	22.9 (41.4)	19.1 (37.7)	25.3 (46.3)

LC50 and LC90 are the concentration of Ag-NPs killed 50% and 90% of individuals, respectively. LCL and UCL are lower confidence and upper confidence limits, respectively.

**Table 2 jof-08-00396-t002:** Field assay of *A. aegypti* larval density with reduction percentages due to treatment with Ag-NPs (10× LC50) fabricated by *A. niger* strain F2.

Treatment	Larval Density with Reduction Percentages (%)							
	before Treatment	Reduction %	after Treatment					
			24 h	Reduction %	48 h	Reduction %	72 h	Reduction %
Ag-NPs (10× LC50)	127.1 ± 4.3 ^a^	0.0	51.4 ± 6.3 ^b^	59.6	32.2 ± 4.1 ^c^	74.7	0.0 ± 0.0 ^d^	100

Data is represented by mean ± SD, different letters in the row are significantly different (*p* < 0.05).

**Table 3 jof-08-00396-t003:** Smoke toxicity of mycosynthesized Ag-NPs-based coils against the adults of *A. aegypti*.

Treatment	Fed Mosquitoes (%)	Unfed Mosquitoes (%)		Total (%)
		Alive	Dead	
Ag-NPs based coil	14.4 ± 1.7 ^b^	22.6 ± 1.9 ^a^	59.0 ± 2.1 ^a^	81.6 ± 0.5 ^b^
Negative control	73.9 ± 0.9 ^a^	23.9 ± 1.6 ^a^	0.0 ± 0.0 ^c^	23.9 ± 1.2 ^c^
Positive control	9.03 ± 1.7 ^c^	37.9 ± 1.7 ^b^	48.2 ± 1.5 ^b^	86.1 ± 1.1 ^a^

The negative control is the coils without any active compounds; the positive control is the Pyrethrin-based coil. Data are represented by means ± SD, and different letters in the same column are significantly different (*p* < 0.05).

## Data Availability

The data presented in this study are available on request from the corresponding author.

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
