# Peer review of "Mycosynthesis, Characterization, and Mosquitocidal Activity of Silver Nanoparticles Fabricated by Aspergillus niger Strain"

_jof, 2022, doi:10.3390/jof8040396_

Round 1

Reviewer 1 Report

In the present article, authors Award et al use a well known and effective technique to silver nanoparticles with a fungal extract to show that they are very effective against the A. aegypti developing stages in laboratory and semi-field conditions. While the experiments are well thought about and performed, but there are some minor and major issues in the MS that need to be addressed before publishing in the journal. I will attach a separate file with my minor comments. My major comments are:

  1. Instead of calling the green-synthesis, use some other alternate terms that is more understandable like environmentally friendly or biorational.
  2. The last paragraph of the introduction has many faulty sentences, please rewrite it.
  3. Section 2.2 (M&M) also need to be rewritten, at present one is not able to understand how the NPs were made, especially if you look at lines 112-113, it makes no sense.Please rectify that.
  4. Similarly there are other areas in the M&M section that would need some correction and refinement so others can follow your methods/protocols to repeat them in needed.
  5. In the experiments performed, I don't see certain controls for the AgNO3 and the fungal culture extract. I see that you discuss that just using the extracts (in case of botanicals) was less effective than the combination with NPs (for other cases) did you test your extract and AG-NPs without the extract that they themselves are not toxic to the organisms (my question originated from this pub: https://doi.org/10.1371/journal.pone.0053186)
  6. whenever referring to a first author and co-athors, you can use et al instead of X ad co-author.
  7. While combining the results and discussion, it is generally a good idea to discuss your results first before going into discussing others work to give context to your work.
  8. LIne 354-357: I don't think this argument works for your case as most known (almost all) resistance is to chemical/synthetic pesticides and no known mechanism exists/known for nanoparticles and other bioratonal control methods. Many other reasons could be attributed for the differences like no proper distribution of the NPs across the containers etc. Either modify the sentences or remove them as you get complete reduction by 72h. (Please feel free to correct me if I am wrong on the resistance part)
  9. In general, has there been any general analyses of NPs toxicity on other invertebrates and vertebrates? If possible discuss, if many of the toxicity are general, then application of these NPs in an open environment could cause the same issue as the traditional methods
  10. Please see the attachment for more detailed information many other corrections in the MS.

Author Response

Dear reviewer, Thank you very much for your valuable comments. We answered All comments point by point. Please see the attachment. 

Reviewer 2 Report

Reviewer comments for Manuscript ID: jof-1663386  Title: Silver Nanoparticles Synthesized by Fungal Strain Aspergillus niger for Biological Control of Aedes aegypti  Note: Generally the concept, design and execution of the study is good, however the minor corrections as stated below are necessary: 1. Pg 1, Line 21- Characterized by color, you don't characterized by color, author should recast the statement  2. In abstract, the first mention of all abbreviations should be written in full 3. Pg3, Line 96- was done by using, 'by' should be expunge  4. Pg3, line110- min should be written in full 5. Pg7, line 226-236- Discussion before results, author should please recast  6. Pg7, line 236-239- Method in the result session, not allowed please 7. Pg12, line 347- 72h - space between figure and unit  8. Pg15, line 459-460- Acknowledgment, however line 454, authors reported no external funding  9. Pg18, line 576- 2016 repeated- please re-check  10. Pg18, line 601- Reference not complete  11. Pg19, line 638- Reference not complete

Author Response

(The authors gave the same response as above.)

Reviewer 3 Report

The manuscript by Mohamed A. Awad et al., deals with the Silver Nanoparticles Synthesized by Fungal Strain Aspergillus niger for Biological Control of Aedes aegypti.

  1. The title should be changed because the fact of using biological material for the synthesis of nanoparticles does not mean that the nanoparticles behave like a living organism. Please review what is the definition of biological control.
  2. I am not sure about the novelty of this manuscript. To my best knowledge, there are a lot of reports about the synthesis of AgNPs from fungi and also their application to control mosquitoes.
  3. Introduction section; authors should review the literature more about the synthesis of AgNPs, their mechanism, why use fungi or another microorganism, etc.
  4. Materials and Methods section

            Fungal used; please provide the reason why used the fungal strain isolated from the archeological manuscript in your study.

Synthesis of Ag-NPs;

  • If the author used NaOH, besides the biomass filtrate, why did not make one control of the synthesis using NaOH and AgNO3 aqueous solution? Because the NaOH plays an important role in the synthesis of the metallic NPs, thus, how to explain that only the biomass filtrate genre AgNPs.
  • The resulting product of the synthesis did not was washed to eliminate residual salts?
  • Explain the recovery process for the pure NPs for use in toxicity experiments.

Toxicity experiments: preparation of the suspension of the Ag-NPs should be detailed.

Smoke toxicity assay; specify the component of the negative control

  1. Results and discussion- All the results were clearly described. However, further discussions are needed, not only the comparison with previous studies but also a deep interpretation of the results. Even more, the importance and scientific meaning of the results.

Synthesis of Ag-NPs;

  • Explain the mechanism of the synthesis of Ag-NPs by Fungi, including NaOH.

Ag-NPs characterization;       

  • It is necessary the FT-IR of the biomass filtrate to compare with the Ag-NPs

Smoke toxicity assay:

  • Mechanism of the insecticidal Ag-NPs as smoke should be interpreted/explained.

Check all the writing, because there is a color difference in the writing (for example; line 20 is black and from line 21 to 20 are greys, after that again from line 30 to 33 is black and after that greys, etc)

Author Response

(The authors gave the same response as above.)

Reviewer 4 Report

The current study focused on the toxicity of Ag-NPs against a disease vector. Furthermore, this nanoparticle was constructed by a fungal strain. The experiment was designed properly and provided very useful data for future study. However, I think the manuscript needs to be improved before it could be published. (1) Line75-77: Nanoparticles are not regarded as eco-friendly and absence of negative impacts on public health compared to chemical insecticides. (2) Line81-83: Helicoverpa armigera and Spodoptera litura are not mosquito vectors. (3) Line109-114: Authors stated that the supernatant (fungal biomass filtrate) was used as a biocatalyst for green synthesis of Ag-NPs. Please provide the proof. (4) Authors should determine the absorbance of Ag-NPs with various concentrations, and provide a standard curve for quantitative analysis. (5) Figure3: the particle size from TEM was bigger than that from DLS? (6) Most experiments are not related to fungi.

Author Response

(The authors gave the same response as above.)

Reviewer 5 Report

jof-1663386-peer-review-v1

Abstract:

This section is well written but lack of conclusions, the authors are requested to add the conclusions of their study

Introduction:

No suggestions look alright

Materials and Methods:

Line 99 please add reference

Line 154 please add reference here

Was the experiments repeated three times independently, or the data was taken from same cups again and again with different intervals (24, 48 and 72 hours), if so how the authors overcome the phenomena of pseudo-replications

Total numbers of cups per replications???

In field experiments six water reservoirs at the Animal House, Institute, Mosquito Laboratory, Faculty of Science, Al-Azhar University, Cairo, Egypt were used, I am confused how 80% humidity at 28 ± 2°C was maintained?????? in water reservoirs, the humidity may be variable?

Data analysis

Well described

Results

Well written

Discussion

The discussion should be separated from results as per format of the JoF

On the bases of above, I strongly believe that this manuscript needs minor revision before publication in JoF

Author Response

To Reviewer #5

Thank you very much for reviewing our manuscript, your agreement, and your valuable comments. We are also grateful for the favorable comments. We made corrections and we hope they meet with your approval. A detailed explanation is given below.

Reviewer comment #: Abstract: This section is well written but lack of conclusions, the authors are requested to add the conclusions of their study

Author response #: Thank you very much for your agreement and your comment. According to instructions for authors, the abstract should be not exceed than 200 word, therefore, we summarized the main results in abstract. However, we adding the main conclusion as follows: “The obtained data confirmed the applicability of biosynthesized Ag-NPs to the biocontrol of A. aegypti at low concentrations.”.   

Reviewer comment #: Introduction: No suggestions look alright

Author response #: Thank you very much for your agreement

Reviewer comment #: Materials and Methods: Line 99 please add reference

Author response #: Done

Reviewer comment #: Line 154 please add reference here

Author response #: Done

Reviewer comment #: Was the experiments repeated three times independently, or the data was taken from same cups again and again with different intervals (24, 48 and 72 hours), if so how the authors overcome the phenomena of pseudo-replications

Author response #: Thank you for your comment. The experiment was repeated three times independently and the result was taken from each replicate at interval times

Reviewer comment #: Total numbers of cups per replications???

Author response #: Three cups for each Ag-NPs concentration.

Reviewer comment #: In field experiments six water reservoirs at the Animal House, Institute, Mosquito Laboratory, Faculty of Science, Al-Azhar University, Cairo, Egypt were used, I am confused how 80% humidity at 28 ± 2°C was maintained?????? in water reservoirs, the humidity may be variable?

Author response #: please accept my apology for this miswritten. we correct the number as 80 ± 5 %

Reviewer comment #: Data analysis: Well described

Author response #: Thank you for your agreement.

Reviewer comment #: Results: Well written

Author response #: Thank you for your agreement.

Reviewer comment #: Discussion: The discussion should be separated from results as per format of the JoF

Author response #: We appreciate the comment of the reviewer, but also the journal accepts the combination between results and discussion, therefore we prepared the manuscript at this form. we will take the reviewer comment in the future study.   

Reviewer comment #: On the bases of above, I strongly believe that this manuscript needs minor revision before publication in JoF

Author response #: Thank you for your positive impacts. All reviewer comments were answered point by point as shown above.

Finally, we hope the response meets the reviewer approval.

Round 2

Reviewer 1 Report

The authors have done a wonderful job of correcting the MS. It is almost ready for publication in the present format. However if time permits and possible, I would recommend to authors to go through the MS once more to make sure there are no minor errors (grammatical or otherwise) within the MS so it reads better to all the international audience. I am writing this since I did notice there were still some minor errors and it is not feasible for me to point out all those in the interest of my time.

Author Response

Thank you very much for reviewing our manuscript, your agreement, and your valuable comments. We are also grateful for the favorable comments. We made corrections and we hope they meet with your approval. A detailed explanation is given below.

Reviewer comment #: The authors have done a wonderful job of correcting the MS. It is almost ready for publication in the present format. However if time permits and possible, I would recommend to authors to go through the MS once more to make sure there are no minor errors (grammatical or otherwise) within the MS so it reads better to all the international audience. I am writing this since I did notice there were still some minor errors and it is not feasible for me to point out all those in the interest of my time.

Author response #: Thank you very much for your agreement. We revised the MS more than one time to correct the minor errors. Also, the English language of the manuscript was subjected to professional editing and proofreading services by “SERVICE SCAPE” for language editing services.    

Finally, we hope the response meets the reviewer approval.

Reviewer 3 Report

The interpreter/explain mechanism of the insecticidal Agnps as smoke still needs improvement. What happens with AgNPs when coils is burned, etc. 

Author Response

Dear reviwer: 

Thank you very much for reviewing our manuscript, your agreement, and your valuable comments. We are also grateful for the favorable comments. We made corrections and we hope they meet with your approval. A detailed explanation is given below.

Reviewer comment #: The interpreter/explain mechanism of the insecticidal Agnps as smoke still needs improvement. What happens with AgNPs when coils is burned, etc. 

Author response #: Thank you for your comment. The possible toxic smoke mechanism was illustrated in the revised MS as follows: “The smoke toxicity of Ag-NPs-based coils can be attributed to their toxic effects on the central nervous system of mosquitos upon exposure to the vapor released from burning coils [74]. Moreover, the vapor released from coil burning can be causing acute irritation to the upper respiratory tracts [75]. The coil burning produces unhealthy air conditions that direct entry to internal organs, ultimately severe organs paralysis, producing smaller larvae, and reducing the fecundity [76].”

Finally, we hope the response meets the reviewer's approval.

Reviewer 4 Report

I have no further comments

Author Response

Dear reviwer:

Thank you very much for your agreement and your approval